

# Continuous mutation of SARS-CoV-2 during migration via three routes at the beginning of the pandemic

Tomokazu Konishi

Graduate school of Bioresource Sciences, Akita Prefectural University, Akita, Japan

## ABSTRACT

**Background**. It remains unclear how severe acute respiratory syndrome coronavirus 2 (SARS-CoV-2) infection started, spread worldwide, and mutated to result in the present variants. This difficulty can be attributed to the limitations associated with the analytical methodology for presenting the differences among genomic sequences. In this study, we critically analysed the early data to explain the start and spread of the pandemic.
**Methods**. Objective analyses of the RNA sequences of earlier variants of SARS-CoV-2 (up to September 1, 2020, available in DDBJ and GISAID) were performed using Principal Component Analysis (PCA). The results were compared with information on the collection dates and location. The PCA was also conducted for 12 variants of interest to the WHO as of September 2021, and compared with earlier data.
**Results**. The pandemic began in Wuhan, China. This strain was suspected to be related to other reported animal viruses; however, they had a minimal similarity. The strain then spreads via three routes while accumulating mutations. Several viral subgroups were identified along the routes, each with a large number of patients reported, indicating high infectivity to humans. These routes were only confirmed by the early data analysis, because newer variants would have more mutations, and would be preferentially be examined by PCA if they were included. On the original axes found in the early variants, the newer variants revealed that they retained previously acquired mutations, which helped to reveal the viral ancestors of the newer variants. The rate of mutation was found to be comparable to that of the influenza H1N1 virus, which causes recurrent seasonal epidemics. Another threat imposed by SARS-CoV-2 is that if the pandemic cannot be contained, new variants may emerge annually, preventing herd immunity.

# INTRODUCTION

More than a year have passed since the start of the coronavirus disease 2019 (COVID-19) outbreak; however, but despite many efforts, the time course of the appearance of this disease is unknown. In most cases, bats serve as coronavirus hosts (*Konishi, 2020b*), but a direct ancestral candidate for this pandemic is uncertain. The virus spread worldwide in a short period of time, producing many variants; however, the entire process remains unclear. To investigate the initial expansion, all data from that period were examined closely on a time course.

Corresponding author
Tomokazu Konishi,
konishi@akita-pu.ac.jp

There are several reasons for problems with this research, but perhaps, the biggest problem lies in the methodology of data analysis, i.e., data clustering. Although phylogenetic clustering has been used to observe the relationships among living organisms (*Yang & Rannala, 2012*), it has both fundamental and practical problems in observing the process of viral evolution, as follows. The principle aspect is that it cannot withstand scientific criticism because it lacks falsifiability (*Ellis & Silk, 2014*). It presents the relationships in the form of a tree. However, recent genomic information reveals that this has not been the right way, as evolution does not take the form a tree; in fact, there are many looping branches since genetic information can be transferred horizontally. For example, viruses may exchange a part of their genomes, and this is called a shift; this would cause a looping (*Konishi, 2020b*). Additionally, estimating the tree shape requires various assumptions that are never verified (*Konishi et al., 2019*; *Yang & Rannala, 2012*). Such a lack of falsifiability renders the application of the phylogenetic tree unsuitable in science (*Ellis & Silk, 2014*). On the other hand, the practical drawback is that it is difficult to see how the tree-shaped relationships correspond to other data set, such as dates. This limitation surely becomes a barrier to identifying virus outbreaks and subsequent transitions.

Recently, *Forster et al. (2020)* found three main clusters in SARS-CoV-2 from 253 sequence samples and estimated the evolution of the virus. It started from one of the clusters, and each of the clusters appeared to have specificities to human races (*Forster et al., 2020*). This study and some other studies used phylogenetic clustering to estimate relationships (*Alm et al., 2020*; *Biswas & Majumder, 2020*; *Gussow et al., 2020*; *Lam et al., 2020*; *Zhou et al., 2020b*). Therefore, the knowledge obtained has little generality and hence cannot be integrated.

Sequence data can be regarded as multivariate data in which each base is an independent variable. Among various methods for multivariate analysis, the principal component analysis (PCA) is acceptable in science as it does not take unverifiable assumptions into account (*Jolliffe, 2002*; *Konishi, 2015*; *Konishi et al., 2019*). Here, 8,974 samples of SARS-CoV-2 virus at the early stage of the pandemic were analysed by PCA. The estimated principal components (PCs) were further compared with the corresponding dates of collection and locations, thereby disclosing how this virus emerged, mutated, and spread.

## MATERIALS & METHODS

### Data and classification

Sequencing data were obtained from the DNA Data Bank of Japan (DDBJ) (*Ogasawara et al., 2020*) and GISAID (*Shu & McCauley, 2017*) databases at 1 Sep 2020; all the data available in the databases at the time were used in the study. Unfortunately, some of the records were rather preliminary and contained sections of uncertain reads as well as extra stretches of repetitions. In order to avoid artifacts that are provoked by the sequencing errors, the corresponding regions were replaced with the average data in the PCA. This treatment practically removes such uncompleted regions from the analysis. The sequences were aligned with DECIPHER (*Wright, 2015*) and manually completed using MEGA (*Kumar et al., 2018*). They were further processed to observe the relationships among samples by

using the direct PCA method (*Konishi et al., 2019*), the recent codes of which are presented in GitHub (*Konishi, 2020a*). The list of used sequences is presented in Figshare (*Konishi, 2021a*). Additionally, 14 early records recovered using deep sequencing data (*Bloom, 2021*) were added. As their collection dates are unknown, the date was assumed to be 23 November 2019.

## PCA

This analysis represents the differences among samples of multivariate data through a set of common directions, which are shown as independent vectors (*Jolliffe, 2002*). All the calculation was performed on R (*R Core Team, 2020*). Here, the samples are mutations that evolved from an original virus. Hence, the samples will fall into several related groups; each group is different from the others, with a unique direction common to the group. The sequence matrix is converted to a stack of Boolean vectors to allow for calculation (*Konishi et al., 2019*). When $m$ is the number of samples (in reality, to cover sequences with $n$ bases, the length of a Boolean vector becomes $5n$; here for simplicity, it is described as below), the matrix $X$ is given as follows:

$$X = \begin{pmatrix} 01_{11} & \cdots & 01_{1n} \\ \vdots & \ddots & \vdots \\ 01_{m1} & \cdots & 01_{mn} \end{pmatrix}.$$

Next, the average of the samples, $a$, is found, and $X$ is centred by subtracting each row with the average: $a = (a_1, a_2, \ldots, a_n)$, $C = X - a$. This centred matrix, $C$, was applied to the PCA. It is subjected to singular value decomposition, $C = U \Sigma V^*$, where $U$ and $V$ are unitary matrices that specify the directions of the differences. As the character of the unitary matrix, the scale of each column and row of $U$ and $V$ is one, $V^*V = I$, and $U^*U = I$. Their columns can be regarded as vectors that represent the axes of each PC. $\Sigma$ is a diagonal matrix that records the scaling of each axis in descending order.

The PCs for the samples, $S$, were found as $S = CV = U\Sigma$. The $CV$ indicates the rotation of $C$ around the centre $a$, retaining the shape of $C$. The results of the rotation are along the axes. This is the same as $U\Sigma$, which is the unitary matrix given the scale. Each of the columns of $S$ represents the PCs: the leftmost column is PC1; the second column is PC2. The descending character of $\Sigma$ orders the scaling of PCs. All of the information is conserved, and all the calculation steps are reversible. Once found, axis $V$ can be applied to other sets of matrices (*Konishi, 2015*). This characteristic is beneficial when applying a classification to newly found samples.

PCs for bases, $B$, can be found in the diagonal direction above, $C^* = V\Sigma U^*$, as $B = C^*U = V\Sigma$. Therefore, $S$ and $B$ are inextricably linked; for example, samples with many positive $B$ in an axis will become highly positive in $S$ along the same axis. On the contrary, the characteristics of a sample that shows a high score on an axis will appear in the same axis as $B$.

To enable comparisons with datasets with different sizes of $n$ or $m$, PCs can be scaled for different sizes (*Konishi, 2015*). The scaled versions of $S$ and $B$, sPC for samples and sPC for bases, are $S/\sqrt{n}$ and $B/\sqrt{m}$, respectively. This is a type of normalization, using which we can directly compare the magnitude of mutations with data from other species, such as influenza.

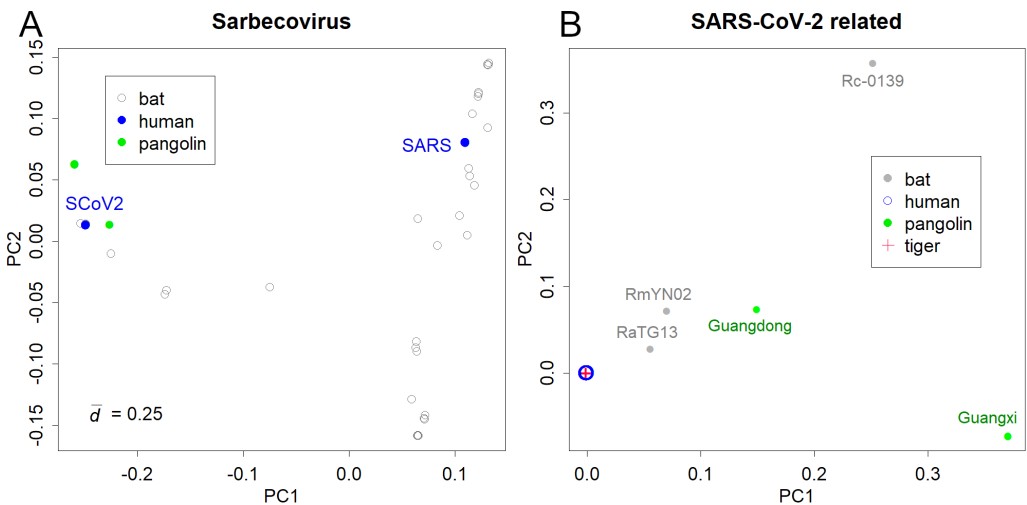

**Figure 1** **PCA for coronavirus.** (A) Sarbecovirus. SARS: SARS-CoV, SCoV2; SARS-CoV-2. The value $d$ is the standard deviation at each base that shows the magnitude of variation. (B) SARS-CoV-2 and related viruses in humans, bats, Malayan pangolins, and a tiger.

The variants of interest to the WHO (*WHO, 2021*) were collected from GISAID (*Shu & McCauley, 2017*) and used in the calculations. In the newer data sets, the mean and unitary matrix V were calculated for these new variants and samples from groups 0–3; two samples out of each group were used for the calculation, to standardize the number of samples in a group. These calculations were applied to all data to obtain the PC. The sets of axes can be downloaded from Figshare (*Konishi, 2020c*).

### Estimation of the magnitude of sample variations

The scale among sample sequences was estimated by mean distances, scaled by the length of sequence m, of virus types. This is a type of standard deviation, $\bar{d} = \sqrt{\sum(x - \bar{x})^2/2mn}$, where $x$, $\bar{x}$, and n are the Boolean of each sample sequence, the sample mean, and the number of samples, respectively. The unit of length is the same as that of the PCA, which will extract the length towards particular directions.

## RESULTS

The PCs of Sarbecoviruses, including SARS-CoV and SARS-CoV-2, is shown in Fig. 1A. The value $\bar{d}$ is the mean standard deviation at the base, indicating the magnitude of variation in the sequence. The presented variations are due to the increasing number of sequence records in bats that have been reported after the SARS outbreak. Six pangolin (*Lam et al., 2020*) and three bat samples (*Murakami et al., 2020*; *Zhou et al., 2020b*) were similar to SARS-CoV-2 (Fig. 1A). The variations in SARS and SARS-CoV-2 were not apparent in the PCA of Sarbecoviruses as a whole.

Bats would be the host of many coronaviruses and pangolins were thought to be the intermediate animal (*Lam et al., 2020*; *Zhou et al., 2020a*); however, the reported sequences were away from the human viruses (Fig. 1B). The PCA totally ignored differences among
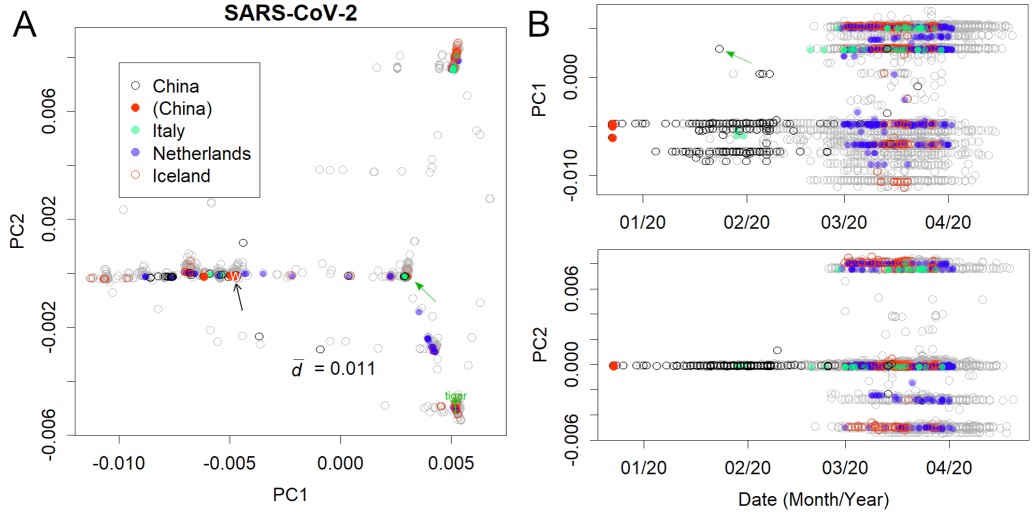

**Figure 2  Human severe acute respiratory syndrome coronavirus 2 (SARS-CoV-2).** (A) PC1 and 2. (B) Principal components (PCs) against date of collection. Some European countries have been coloured to clarify the spread of the virus. Italy and Iceland rank among the early and late spread countries, and the Netherlands is in between. Plural variants are found in each country, suggesting multiple infection routes. Black and green arrows present the first variants in Wuhan and Europe, respectively.

human samples; instead, it detected the large differences between humans and the other hosts (list of the sequences is available at Supporting Information). In fact, the pangolin virus and bat virus differed from severe acute respiratory syndrome coronavirus 2 (SARS-CoV-2) by 3–5,000 bases and 1–2,000 bases, respectively. In contrast, variations in SARS-CoV-2 were very limited. For example, the most distant sample pair in sPC1 differed by only 17 bases. The standard deviation of the sample differences was 3.3 bases. The total ignorance of differences among human samples in the PC1 and PC2 shows that the directions of human virus mutations do not correspond to the differences between the hosts. In contrast, the sample of a tiger was located inside the human samples.

The analysis further focused on SARS-CoV-2 in humans (Fig. 2); the axes were found only using human viruses and excluding those from pangolins, tigers, and bats. A complete list of PC is in Figshare (*Konishi, 2021a*). The virus mutated along three routes until April 2020 (Fig. 2A; the contribution of PCA is presented in Fig. S1). The first reports started in Wuhan, China (Fig. 2B, black arrow). The recovered records were located in similar positions on the PCA (red dots). Then the samples diffused horizontally in PC1. A variety that occurred in China transferred to Europe (Fig. 2 and Fig. S2, green arrow) and separated into two routes, divided along the PC2 axis. A new variety would be established when a small number of people carrying a mutant move to a new location and the infection spreads further. The lines that appeared on PCA reflected the trajectory of virus migration; accumulating mutations further elongated along the routes. Since only limited countries reported sequences with varied numbers (Table S1), the width of the lines would become thick or thin in panel A, and show horizontal lines in panel B. The similarity of the varieties found in Australia, England, Taiwan, and the United States (Fig. S3) is probably due to
the fact that each country trades with many other countries. It should be noted that the locations are critical for the spreading route and not for the human race.

The mutation rates of PC1 and PC2 (Fig. 2) were found to be reasonably high. Although the mutation rate of seasonal human coronaviruses is rather slow (*Konishi, 2020b*), the mutation rate of SARS-CoV-2 does not follow that trend. Within a few months, the coronavirus had changed by 0.01 in PC1, which is a speed comparable to or even faster than that of the H1N1 influenza virus, which mainly mutates via a single route (Fig. S4) (*Konishi, 2019*).

Additionally, the Pango lineage of the first samples was B according to the new classification system (*Rambaut et al., 2020*). The variant that went down PC1 was A, and the one that went up was B.1, which is called group 0 (Fig. 3A). This group1 spread to Europe, where it went up and down PC2, to form groups 2 and 1, respectively. The new classification system does not distinguish between them; this is probably because the system is defined with reference to the phylogenetic tree based on distance.

The original axes that were found in the early sets of human viruses separated the data into four main groups, which were tentatively numbered from 0 to 3 (Fig. 3A). The first group that was formed in Europe was designated as 0, which is located near the origin. In PC2, the group that moved to a large negative value was designated as 1, whereas the group that moved to a positive value was designated as 2. The group located in the negative region in PC1 was designated as group 3, which includes the original strain. However, it has already disappeared (Fig. 2B), which may be due to its weaker infectivity. Additionally, several subclass clusters appeared because each group had large numbers of patients, thus proving that the virus is highly infectious to humans.

The routes through which the initial infection had spread, appeared only when the SVD calculations were performed using relatively early data (including the first period) (Figs. 2 and 3A). The newer variants of interest to the WHO (*WHO, 2021*) could be represented on the original human axes (Fig. 3A) because these appeared in subgroups that had been previously recorded. In particular, all variants, except for $\eta$ and $\mu$, were included in the three subgroups.

If we calculate the PCA using newer variants, the early variants will be clustered around the center, and the routes will not be visible even in the lower-level axes (Figs. 3B–3D). Because the new variants have more mutations, they are given more weight. The scale of the new axes was much larger than that of the earlier axes (compare Figs. 3A and 3B–3D), and PCA preferentially picked up the features of the new variants using multiple axes. This indicates that the mutations in the new variants are larger and occur at different positions in the genome. Even variants classified under the same Pango lineage do not necessarily form a single cluster, as mutations continue to occur, as seen in the $\lambda$ and $\theta$ variants (Fig. 3B). Some variants belong to the same cluster as the older ones, but their features are displayed in the lower PCs (Figs. 3C and 3D).

## DISCUSSION

The origin of the variants would be at least similar to the first samples. The first samples, including the recovered records, were located near the PCA centre (0, 0), which is the

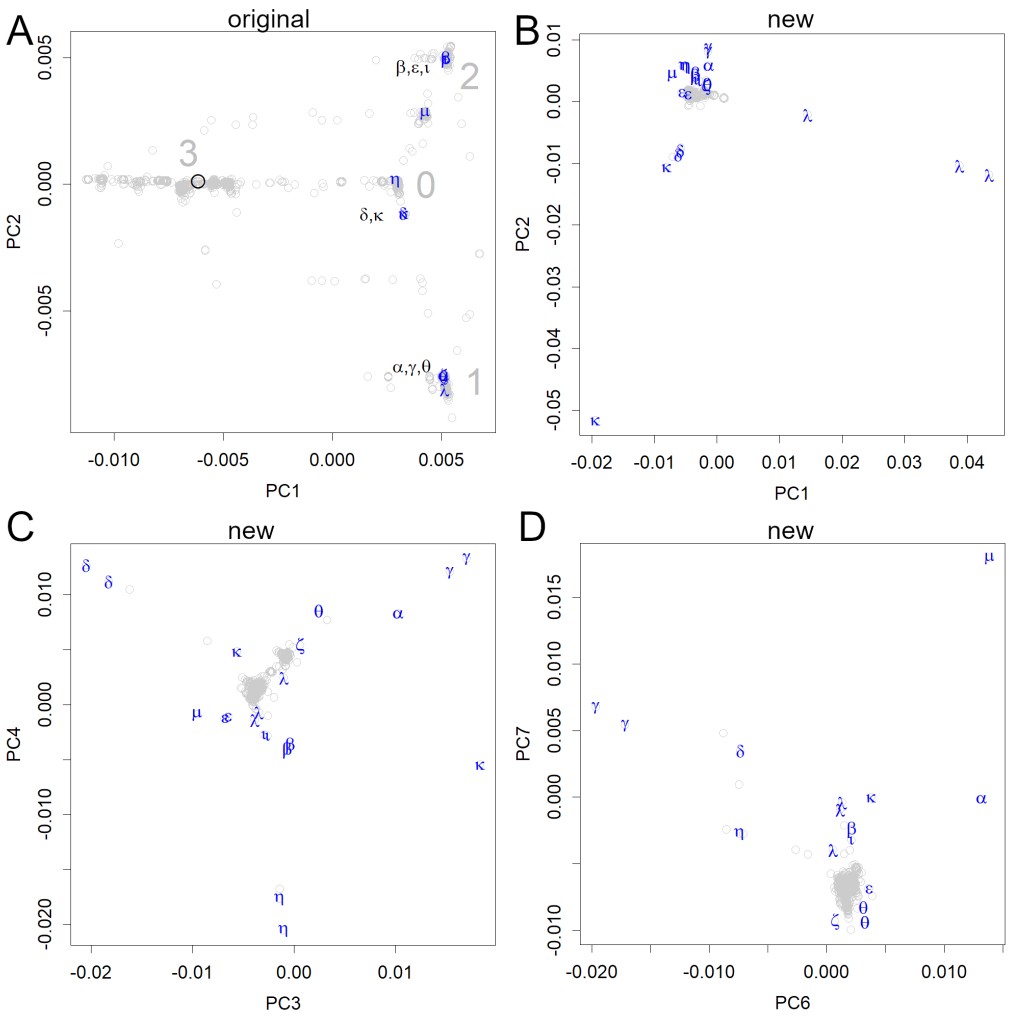

**Figure 3  PCA locations of variants of interest to the WHO.** (A) The variants are shown in the axes obtained from the original strain; the sequence data were multiplied with the unitary matrix *V* obtained by SVD. The variants appeared in groups 0, 1, and 2; many of them belonged to three small groups (the variants, Pango lineages, and earliest documented samples). α: alpha, B.1.1.7, England; β: beta, B.1.351, South Africa; γ: gamma, P.1, Brazil; δ: delta, B.1.617.2, India; ε: epsilon, B.1.427&429, USA; ζ: zeta, P.2, Brazil; η: eta, B.1.525, multiple countries; θ: theta, P.3, the Philippines, ι: iota, B.1.526, USA; κ: kappa, B.1.617.1, India; λ: lambda, C.37, Peru, μ: mu, B.1.621, Colombia. (B–D) All data were shown on the axes obtained from the new variants. Each axis shows the features of some variants but does not reveal the initial infection routes.

data average. According to the central limiting theorem, if a strain continuously mutates, it should appear near the data average among the formed variations. The first samples showed slightly negative values in PC1, but this bias may be attributed to a large number of reports from Europe. The human infection of a bat virus, RaTG13 (Fig. 1B), in 2012 was recently reported (*Arbuthnott, Calvert & Sherwell, 2020*). However, if the present coronavirus pandemic started from individuals infected by RaTG13, many derivatives should be generated from the strain soon. These would become similar to RaTG13, and

at least a part of them should remain among humans and should appear surrounding RaTG13, rather than the first Wuhan variant, on the PCs. Hence, the containment of RaTG13 in humans seems to be successful; accordingly, neither this nor the pangolin virus was the origin of the virus responsible for the pandemic. Similarly, a large sample review reported infection in the United States in early 2020 (*Basavaraju et al., 2020*). However, if it were a type with a different sequence from the first samples, their mutation process should remain on the PC. Since the process by which the first samples mutated is clear, it is unlikely that a different side-stream disappeared.

The mutation rate of SARS-CoV-2 was high. The presence of three routes for mutation resulted in a rapid expansion of the differences among variants. In European countries such as Iceland, several varieties have been found (Fig. 2A). Accumulation of these mutations demand novel immunities for virus control and hence, weaken the effect of herd immunity. The virus is bound to repeat seasonal epidemics such as influenza.

The results presented here are quite different from those reported by Forster et al. (*Forster et al., 2020*). One of the reasons is in their decision that the bat sample is the direct origin of SARS-CoV-2; however, the appropriateness of this decision was not verified. This relates to the most basal unverifiable assumption of the clustering methods, whether the samples should be directly connected or not. Additionally, the previous study did not take the passage of time into account. Another disadvantage of a phylogenetic tree is that it is more difficult to reconcile with other information. In contrast, PCA generates a spreadsheet with the same dimension as the sequence matrix, facilitating data integration. Actually, the estimation of the origin of the outbreak in this report has two distinct roots of evidence: the close proximity to the mean data and the start of the records.

The PCA finds bases common to a particular group and scores them. Therefore, all processes of mutations will be treated separately. Accordingly, the genealogy of mutation becomes clear naturally, and the three routes described here have come into view. This is a big difference from the distance-based phylogenetic tree method: the distance loses base information. Although the new system (*Rambaut et al., 2020*) failed to detect differences between the classes, the groups presented here prevailed until June 2021. In Fig. 2, we used data from the beginning of the outbreak to September 1, 2020, to find the original axes of the earlier sets of human viruses, which helped in revealing the initial transmission and mutation. Thereafter, we identified four groups of variants in that period that were particularly infectious (Fig. 3A). These variants probably had a higher affinity for humans than the original variants did, and the acquired mutations conferring this advantage tend to be conserved. Indeed, when the newer variants were presented, they showed values identical to those of the previous subgroups (Fig. 3A). Therefore, the positions of the new variants would reveal their ancestors. Unlike the Pango lineage classifications (*Rambaut et al., 2020*), which are based on cluster analysis, the results presented here would relate more to the evolution of the virus. In comparison with clustering, PCA would be more desirable to observe the classification, since the viral evolution history becomes available.

However, the PCA is sensitive to noise and bias. Newer variants have more mutations and a larger number of patients, adding uneven weights that could add bias to PCA. The original human axes had been found in all initial data, but it was possible because there was

no strong infectious strain at that time. If the present data are not standardized, the axes will be dominated by strong variants, such as α and δ. The new axis presented in this study is based on such considerations, but if the newer variants are omitted in the standardization process, they will be missed out on the analysis. The new axes would have a high sensitivity to only the variants that were used in the SVD. In order to discover new variants, biased results are inevitable as all data need to be used. Also, it has to search through the lower level axes.

The μ variants seem to have completely different mutations from the others, with PC6 and PC7 showing unique positions (Fig. 3D). Inherently, these unique mutations can affect the efficacy of the vaccine; moreover, there is concern that this strain will cause breakthrough infections (*Uriu et al., 2021*) This is also true for strains such as λ and κ. As the vaccinated population increases, a new selective pressure to break through the acquired immunity seems to arise. This is a different selective pressure than that faced by α, which had been based on a higher human infection rate. Periodic updates to vaccines are also necessary.

Investigation of the parent virus of SARS-CoV-2 would be difficult. We will not think of the tiger as the intermediate animal, even if the variant is exactly the same as that in humans (Figs. 1 and 2A). In fact, the data were far from the mean data, and the collection date was too recent. However, what if a likely virus was found in bats? It is unnatural to estimate that the bat or pangolin viruses could be the ancestral strain of SARS-CoV-2; rather, they may belong to a subclass that includes SARS-CoV-2 and the undiscovered ancestor. We may know only a part of the variations of bat and other animal viruses. In fact, it seems that wild pangolins are not good hosts for coronaviruses (*Lee et al., 2020*). It is likely that these pangolins were infected during their delivery to the market.

Investigation of the parent virus carries a high risk due to its experimental nature as any leak could cause a new pandemic. The virus could have been infected by a human or attributed to laboratory technical errors, such as contamination from human samples. It is certain that there is a subclass of coronavirus that infect humans, pangolins, as well as bats, and this subclass includes SARS-CoV-2 (Fig. 1A) (*Konishi, 2020b*). They would have similar sequences; however, amplification and sequencing of an unknown variant would require new sets of primers. Hence, it is beneficial to store variants by maintaining the virus in cultured cells (*Matsuyama et al., 2020*) or other hosts. However, this involves the risk of accidental leaks of viruses that have been acclimated to the origin of cells, usually primates. It should be noted that even a single accident can result in a new pandemic. There is another approach; some studies have used massive parallel sequencers to obtain a mixture of samples, which may contain multiple viruses (*Lam et al., 2020*; *Zhou et al., 2020b*). Although the approach is interesting, it may not be suitable for identifying a novel strain.

## CONCLUSIONS

Using PCA, it became clear that the variants mutated through three routes and spread. The first variants immediately began to mutate, and these mutations formed the PC1. Shortly

after crossing Europe, mutations in another direction occurred, creating the PC2. The variants that moved negatively on PC1 were the first variants to have disappeared by April 2020, after which three groups remained. Mutations will continue, but they will occur within these three groups because new mutations will be recorded on newer axes. The superiority of PCA is clear because the strains, up to this point, could hardly be classified using the phylogenetic tree method. Additionally, it can reveal the direct relationships between various variants with respect to viral evolution.

## ACKNOWLEDGEMENTS

We would like to thank Editage and Enago for English language editing.

### Funding

The authors received no funding for this work.

### Competing Interests

The authors declare there are no competing interests.

### Author Contributions

- Tomokazu Konishi conceived and designed the experiments, performed the experiments, analyzed the data, prepared figures and/or tables, authored or reviewed drafts of the paper, and approved the final draft.

### Data Availability

The data is available at Figshare: Konishi, Tomokazu (2020): Continuous mutation of SARS-CoV-2 during migration via three routes. figshare. Journal contribution. https://doi.org/10.6084/m9.figshare.12365777.v2.

### Supplemental Information

Supplemental information for this article can be found online at http://dx.doi.org/10.7717/peerj.12681#supplemental-information.

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
