# Peer review of "Continuous mutation of SARS-CoV-2 during migration via three routes at the beginning of the pandemic"

_PeerJ, doi:10.7717/peerj.12681_

## Round 0.1 · original submission · Major Revisions

Dear Dr. Konishi:

Thanks for submitting your manuscript to PeerJ. I have now received two independent reviews of your work, and as you will see, the reviewers raised some concerns about the research (mostly the dated nature of the dataset). Despite this, these reviewers are optimistic about your work and the potential impact it will have on research studying Sars-CoV-2 distribution. Thus, I encourage you to revise your manuscript, accordingly, taking into account all of the concerns raised by both reviewers.

Aside from the other concerns of the reviewers, please consider that the alpha and delta variants have replaced most of the older viruses across the world. Thus, your analysis has lost its original merit. However, if you can update the data until a recent period (e.g., Aug/2021), which is a major task -- the paper might add additional scientific merit to the existing knowledge of SARS-CoV-2 and its evolution.

I look forward to seeing your revision, and thanks again for submitting your work to PeerJ.

Good luck with your revision,

-joe

Reviewer 1 ·

Basic reporting

The paper i well written and easy to read with som clear messages. The number of new publications within the field of COVID-19 is overwhelming and it can be difficult to keep and overview. However, the author gives a good and sufficient introduction to the subject.

Experimental design

The origin and spread of SARS-CoV2 and the emerging of new variants is a very important subject. In this paper the author analysed 8.974 sequences of early samples of SARS-CoV-2, using principal component analysis (PCA) instead of using phylogenetic threes.
The mathematical background for PCA is hard to understand for a clinician, but PCA is a validated and recognized method.
The sequence data were obtained from the DNA Data Bank of Japan and GISAID. It is not stated whether the 8.974 samples represents all sequences in the data-bases or if not, how they were selected.

Validity of the findings

Data shows that bats and pangolins are not the intermediate animal for the origin of SARS-CoV-2. Further it spread and mutated through three routes.
One of the main conclusion is that the rate of mutation for SARS-CoV-2 is comparable to Influenza A H1N1. However, the mutation rate is only mentioned in the discussion and not in results.
The data and conclusions are valid.

Additional comments

This paper add to our knowledge about the origin and spread of SARS-CoV-2. It has som clear and important messages.

Suggest that the first sentence in the introduction is changed form: Years have passed since... to More than a year have passed since...

Reviewer 2 ·

Basic reporting

Tomokazu Konishi presented principal component analysis (PCA) of SARS-CoV-2 from DNA data bank of Japan and GISAID. The author collected data up until 1 Sept 2020 and claimed some benefits of PCA over phylogenetic based analysis. However, I have some major concern on authors assumption and analysis
1) Pangolin CoV: Articles published by Jimmy Lee (Lee, J., Hughes, T., Lee, MH. et al. No Evidence of Coronaviruses or Other Potentially Zoonotic Viruses in Sunda pangolins (Manis javanica) Entering the Wildlife Trade via Malaysia. EcoHealth 17, 406–418 (2020). https://doi.org/10.1007/s10393-020-01503-x) showed that specimens collected from naturally occurring Pangolins were negative for Coronavirus at family level. Thus detection of CoV from Pangolins seems like contamination through the wild animal marketing chain. Thus, this is unlikely that Pangolin played any key role in the origin of SARS-CoV-2. I would suggest the author consider this point and reanalyze the PCA by excluding the sequences reported from pangolins and then make a comparative analysis (including pangolin CoV and excluding Pangolin CoV).
2) Tiger CoV: The sequences reported from Tiger were almost identical to human. The most likely explanation is that the tiger received the virus from a human and there is no modification of the virus in the Tiger. Thus, Tiger might be a very poor host for SARS-CoV-2 in terms of viral mutation.
3) H1N1 influenza virus (author mentioned N1H1 at line 200): The purpose such comparison is not clear to me. Not enough data was presented on the H1N1 virus.
4) Data is too old: The dynamic if SARS-CoV-2 is changing very rapidly – the analysis using data up until Sept/2020 seems too old now. Most of those viruses is now replaced with delta and other VOC. Please update the data and re-analyze them. Write the discussion relevant with recent VOC including alpha, beta, delta, lambda variants.
5) Abstract: The abstract is written poorly, please mention the source of data and duration of data collection in the method of Abstract. Write the raw findings in the results section of the Abstract.

Experimental design

NA

Validity of the findings

See the comments above

Additional comments

NA

---

## Round 0.2 · accepted · Accept

Dear Dr. Konishi:

Thanks for revising your manuscript based on the concerns raised by the reviewers. I now believe that your manuscript is suitable for publication. Congratulations! I look forward to seeing this work in print, and I anticipate it being an important resource for groups studying Sars-CoV-2 evolution and dissemination. Thanks again for choosing PeerJ to publish such important work.

Best,

-joe

Reviewer 2 ·

Basic reporting

The author has addressed my comments adequately! This looks a better draft now.

Experimental design

NA

Validity of the findings

With the acknowledgment of Pangolin's CoV and tiger's CoV -- the paper has a clear communication now.

Additional comments

Well done and happy to accpet!